# Effects of portable air cleaners and A/C unit fans on classroom concentrations of particulate matter in a non-urban elementary school

Alexandra Azevedo[1☯‡], Jahred Liddie[1☯‡], Jason Liu[1☯‡], Jessica E. Schiff[1☯‡]*, Gary Adamkiewicz[1☯‡], Jaime E. Hart[1,2☯‡]

1 Department of Environmental Health, Harvard T.H. Chan School of Public Health, Boston, Massachusetts, United States of America, 2 Channing Division of Network Medicine, Department of Medicine, Brigham and Women's Hospital, Harvard Medical School, Boston, Massachusetts, United States of America

☯ These authors contributed equally to this work.
‡ AA, JL, JL and JES share first authorship on this work. GA and JEH shared senior authorship on this work.
* jschiff@hsph.harvard.edu

**Data Availability Statement:** All relevant data are within the paper and its Supporting information files.

## Abstract

Given the increased use of air cleaners as a prevention measure in classrooms during the COVID-19 pandemic, this study aimed to investigate the effects of portable air cleaners with HEPA filters and window A/C fans on real-time (1 minute) concentrations of PM less than 2.5 microns ($PM_{2.5}$) or less than 1 microns ($PM_{1.0}$) in two classrooms in a non-urban elementary school in Rhode Island. For half of each school day, settings were randomized to "high" or "low" for the air cleaner and "on" or "off" for the fan. Descriptive statistics and linear mixed models were used to evaluate the impacts of each set of conditions on $PM_{2.5}$ and $PM_{1.0}$ concentrations. The mean half-day concentrations ranged from 3.4–4.1 µg/m$^3$ for $PM_{2.5}$ and 3.4–3.9 µg/m$^3$ for $PM_{1.0}$. On average, use of the fan when the air cleaner was on the low setting decreased $PM_{2.5}$ by 0.53 µg/m$^3$ [95% CI: -0.64, -0.42] and use of the filter on high (compared to low) when the fan was off decreased $PM_{2.5}$ by 0.10 µg/m$^3$ [95% CI: -0.20, 0.005]. For $PM_{1.0}$, use of the fan when the air cleaner was on low decreased concentrations by 0.18 µg/m$^3$ [95% CI: -0.36, -0.01] and use of the filter on high (compared to low) when the fan was off decreased concentrations by 0.38 µg/m$^3$ [95% CI: -0.55, -0.21]. In general, simultaneous use of the fan and filter on high did not result in additional decreases in PM concentrations compared to the simple addition of each appliance's individual effect estimates. Our study suggests that concurrent or separate use of an A/C fan and air cleaner in non-urban classrooms with low background PM may reduce classroom PM concentrations.

## Introduction

Airborne transmission of the novel respiratory virus SARS-CoV-2 has prompted increased scrutiny of indoor air quality (IAQ), especially in settings where many people gather such as

**Funding:** Jahred Liddie, an author on this project, was supported by a Training Grant in Environmental Epidemiology (T32 E007069) from the National Institute of Environmental Health Sciences. Funds from the Department of Environmental Health, Harvard TH Chan School of Public Health and the Harvard Chan NIEHS Center (NIH/NIEHS P30 ES000002) were used to purchase and maintain the equipment used for the study. The funders had no role in study design, data collection and analysis, decision to publish, or preparation of the manuscript.

**Competing interests:** The authors have declared that no competing interests exist.

schools, offices, and restaurants [1–3]. To further improve indoor air quality and make indoor environments healthier, the Biden Administration launched an effort in March 2022 to improve ventilation and reduce the spread of COVID-19 in buildings, including school buildings [4]. In schools, where attendance is typically mandatory and classrooms can reach a high occupant density, environmental health experts recommend advanced air ventilation and filtration to protect students and teachers [5]. While it is possible to achieve high ventilation rates in classrooms by opening doors and windows, filtration presents a set of challenges. Many schools do not have windows that can be opened for safety or energy efficiency reasons. Heating, ventilation, and air-conditioning (HVAC) systems in older schools are typically equipped with low-grade filters such as minimum efficiency reporting value (MERV) 8, which captures approximately 84% of PM 3–10 μm in diameter, 20% of PM 1–3 μm in diameter, and is not rated for capture efficiency of particles smaller than 1.0 μm [6]. The Center for Disease Control and Prevention (CDC) recommends upgrading to more efficient filters such as MERV 13 and higher, especially in buildings that recirculate air within the same room or same local ventilation zone [6, 7]. However, the mechanical ventilation systems in many school districts are too old to be compatible with newer filters [8, 9]. For school districts that are structurally capable of adding MERV 13 filters, costs associated with installation, operations, and maintenance can easily exceed millions of dollars [9]. Given the challenges of installing new HVAC systems or filters in school buildings, portable air cleaners and fans have become popular, less expensive alternatives to reduce PM concentrations. Air cleaners with high-efficiency particulate air (HEPA) filters, which are at least 99.97% efficient at capturing particles 0.3 μm and larger in size, offer the possibility of substantially reducing SARS-CoV-2 viral particles as well as improving overall IAQ [6, 8].

PM is one of the most common pollutants that could potentially degrade air quality in classrooms [10]. Indoor PM levels are influenced by several factors, such as ambient air pollution levels, air exchange rates, occupancy, type and intensity of indoor activities, and particle sizes [10–12]. One study in Munich, Germany found that PM concentrations in classrooms are about six times higher than outdoor concentrations [13]. Children are particularly vulnerable to potential health consequences related to PM exposure due to their immature respiratory and immune systems and greater breathing rates per body weight [16]. Beyond the current context of the COVID-19 pandemic, improved air ventilation is associated with lower school absenteeism, better performance on cognitive function tests, and fewer respiratory symptoms, such as those related to asthma, lung inflammation, and allergies [14–16]. $PM_{2.5}$ exposure in children has been associated with asthma incidence (OR = 1.10, 95% CI:1.01, 1.20), prevalence of asthma symptoms (OR = 1.08, 95% CI:1.02, 1.16), and rhinitis (OR = 1.15, 95% CI: 1.05,1.26) [17]. A review study of 33 articles further provided evidence that exposure to particulate air pollution has adverse impacts on children's respiratory health, with stronger negative effects seen among children with asthma [18].

While portable air cleaners are commonly used to reduce PM concentrations, especially in locations near roadways and other sources of air pollution, less is known about the use of conventional fans. Several studies have tested the use of portable air cleaners in school settings, characterizing their effectiveness in reducing exposure to $PM_{2.5}$ alone [19]; $PM_{10}$ alone [17]; $PM_{2.5}$ and $PM_{10}$ [12, 20]; airborne allergens [21]; ultrafine particles (UFPs), black carbon, $PM_{2.5}$ and $PM_{10}$, and volatile organic compounds (VOCs) [16]; as well as their effect on indoor $NO_2$ and $CO_2$ concentrations [22]. Only one study has evaluated the performance of fans in mitigating PM levels in classrooms [23]. The objective of this study is to provide data on the effectiveness of portable air cleaners and fans operating in tandem to reduce $PM_{2.5}$ and $PM_{1.0}$ concentrations in occupied classrooms.

## Materials and methods

### Study location

The study was conducted in a public elementary school in a small town in Rhode Island, USA. Two classrooms of similar size, occupancy, age of students, and ventilation conditions were selected for air monitoring. Both classrooms were rectangular, with areas of approximately 140 $m^2$ and 152 $m^2$ (volume 427 $m^3$ and 463 $m^3$) for classroom A and B, respectively (S1 Fig). Neither classroom had windows that open, except for the windows in which the A/C units were located. Classroom A was located on the first floor and classroom B was located on the second floor of the same building.

Prior to the start of the study, each classroom was equipped with a portable air cleaner with an H13-grade True HEPA filter (Medify Air Model MA-112) and a window A/C unit (LG 15000 BTU Model LW1516ER) with separate cooling and fan functions. The volume flow through the air cleaner can be adjusted in five increasing settings: "sleep," 1, 2, 3 and 4. According to the manufacturer, the air cleaner has a reported maximum clean air delivery rate (CADR) rating of 950 $m^3$/h and can cover an area of 762 $m^2$ with two air changes/h. While it is recommended that HEPA filters be cleaned regularly and replaced every six months, the school's maintenance schedule was unknown. However, as the air cleaners had been installed after the beginning of the COVD-19 pandemic, we can assume that they were under six months old at the time of sampling. The fan function on the A/C unit could be adjusted in three increasing settings: 1, 2, and 3. Due to school policies, the air cleaners were operated continuously during the school week (24 h, M-F) throughout the study period, while the A/C fans were operated only during school hours (8:30am to 3:45pm, M-F). The A/C cooling function was not used during the study period.

### Sampling methods

Two personal aerosol monitors (SidePak Model AM510, TSI Inc., Shoreview, MN, USA), one measuring $PM_{2.5}$ and one measuring $PM_{1.0}$, were placed side-by-side across the room (approximately 12 m away) from the air cleaner and A/C unit in each of the classrooms. The same units were used in the same classrooms throughout the sampling period. The inlets of the aerosol monitors were located about 1–1.2 m above ground level. Additionally, one HOBO data logger (HOBO U12 Temp/RH/Light/External Data Logger, Onset Computer Inc., Pocasset, MA, USA) was placed next to the aerosol monitors in each classroom to measure temperature and relative humidity.

Samples were collected at one-minute intervals during the school week (M-F) from 3/31/2021 to 4/16/2021, for a total of eleven days. Different sets of conditions were used in each classroom each half-day. The HEPA cleaner settings ("high" = setting 4 or "low" = setting 1) and fans ("fan on" = setting 1 or "fan off" = 0) for classroom A and classroom B were assigned randomly prior to the start of sampling using R version 4.1.2 [24]. The schedules of half-day settings were distributed to teachers at the start of sampling, as well as log sheets to record deviations from the schedule and times of events that may impact PM concentrations (e.g., bus arrivals, times during which all students typically leave the classroom). We defined noon (12pm) as the cutoff for each trial given that teachers typically transferred from one trial to the next between noon and 12:30pm on each day. Temperature and relative humidity (RH) for each classroom, measured at one-minute intervals, were matched to the PM data.

### Quality assurance and quality control procedures

The SidePak aerosol monitors were factory-calibrated to the respirable fraction of standard ISO 12103–1, A1 Test Dust prior to the start of data collection. Prior to data collection, all

monitors were co-located within a controlled environment and operated for several hours to ensure all monitors functioned properly. The monitors were deemed to have an acceptable level of precision if the average percent difference in PM concentrations between the co-located instruments was ≤ 10%. If this threshold was not met, correction factors were calculated to ensure that differences in the collected data were not due to differences in the instruments used. Exceedances of this threshold were reviewed within the context of testing data, given that greater percent error was possible for these monitors at lower concentrations.

## Statistical analysis

The raw data were trimmed to standard-length school days (8:30am to 3:45pm) and data pertaining to days on which there was no school (i.e., weekends and holidays) were removed. Summary statistics, box plots, and histograms were first generated for $PM_{2.5}$ and $PM_{1.0}$ at the half-day and minute level. With our dataset of half-day trial averages, we utilized one-way analysis of variance (ANOVA) to determine if there were significant differences between trial averages.

We also developed linear mixed models using concentration data at the one-minute level. We defined three fixed effects: fan (1 = on, 0 = off), air cleaner (1 = high, 0 = low), and fan*air cleaner, and one random effect: half-day (1 = morning, 0 = afternoon). We included a fixed effect for classroom (0 = classroom B (second floor), 1 = classroom A (first floor)) to control for time-invariant differences between the two classrooms. Given the potential for high autocorrelation in minute-level data, we also used a covariance structure following autoregressive 1 (AR 1) in the linear mixed models. Given large changes in relative humidity over our study period, a secondary analysis considered relative humidity in each classroom.

All descriptive statistics and analyses were performed using R. Statistical significance was set using α = 0.05.

## Results

A total of 12 half-day trials of *fan off / air cleaner low* and *fan off / air cleaner high*; and 11 half-day trials of *fan on / air cleaner low* and *fan on / air cleaner high* were completed in the classrooms. We removed one full day of sampling for one classroom due to deviations from the scheduled settings. Given the observed percent differences in the pre-sampling data, correction factors were applied to all measurements from both monitors in one of the classrooms. The mean half-day concentrations ranged from 3.4–4.1 μg/m$^2$ for $PM_{2.5}$ and 3.4–3.9 μg/m$^2$ for $PM_{1.0}$ (Table 1). Mean temperatures were similar across days, with temperatures ranging from 69.6–70.9 ℉. However, mean relative humidities varied more widely, with a range of 31.5–37.6%. The lowest mean relative humidity was observed in the *fan on / air cleaner high* trials, while the greatest was observed in the *fan off / air cleaner low* trials.

No significant differences between half-day averaged $PM_{2.5}$ and $PM_{1.0}$ concentrations were observed between the four trials (Table 2). The linear mixed model results are shown in Table 3. The differences in *N* included are due to missing humidity information. When the air cleaner was on low, use of the fan was associated with an average 0.53 μg/m$^3$ [95% CI: -0.64,-0.42] decrease in $PM_{2.5}$ concentrations compared to having the fan off. When the fan was off, use of the air cleaner on a high setting decreased concentrations, on average, by 0.10 μg/m$^3$ [95% CI: -0.20, 0.005] compared to low setting. For $PM_{1.0,}$ when the air cleaner was on low, use of the fan decreased concentrations, on average, by 0.18 μg/m$^3$ [95% CI: -0.36, -0.01]. When the fan was off, use of the air cleaner on the high setting decreased $PM_{1.0}$ concentrations, on average, by 0.38 μg/m$^3$ [95% CI: -0.55, -0.21] compared to the low setting. The product terms for fan on * air cleaner high were statistically significant and positive for $PM_{2.5}$ and

**Table 1. Distributions of PM2.5, PM1.0, temperature and relative humidity under different scenarios of HEPA cleaner and air conditioning unit fan use in elementary school classrooms in Rhode Island, USA.**

| | | Fan off, filter low | | Fan off, filter high | | Fan on, filter low | | Fan on, filter high | |
|---|---|---|---|---|---|---|---|---|---|
| | | Half-day | Minute-level | Half-day | Minute-level | Half-day | Minute-level | Half-day | Minute-level |
| PM2.5 | Concentration [$\mu g/m3$] | | | | | | | | |
| | Min | 2.6 | 1.0 | 2.0 | 1.0 | 2.0 | 1.0 | 2.5 | 2.0 |
| | Median | 3.8 | 4.0 | 3.9 | 4.0 | 3.6 | 3.62 | 3.3 | 3.6 |
| | Mean ± SD | 4.1 ± 0.9 | 4.0 ± 1.7 | 3.7 ± 1.1 | 3.8 ± 1.4 | 3.4 ± 0.8 | 3.4 ± 1.2 | 3.7 ± 0.8 | 3.7 ± 1.1 |
| | Max | 5.6 | 53 | 5.8 | 29 | 4.4 | 20 | 5.1 | 18 |
| | N | 12 | 2637 | 12 | 2625 | 11 | 2370 | 11 | 2338 |
| PM1.0 | Concentration [$\mu g/m3$] | | | | | | | | |
| | Min | 2.5 | 1.6 | 2.3 | 0.5 | 2.1 | 1.03 | 2.3 | 2.0 |
| | Median | 3.4 | 3.0 | 3.8 | 3.86 | 3.5 | 3.35 | 3.2 | 3.1 |
| | Mean ± SD | 3.9 ± 1.6 | 3.9 ± 1.7 | 3.9 ± 1.1 | 4.0 ± 1.3 | 3.8 ± 1.5 | 3.8 ± 1.7 | 3.3 ± 0.8 | 3.3 ± 0.9 |
| | Max | 7.5 | 27 | 5.9 | 17.8 | 6.7 | 33 | 4.7 | 9.0 |
| | N | 12 | 2637 | 12 | 2432 | 11 | 2313 | 11 | 2337 |
| | Temperature [F] | | | | | | | | |
| | Mean ± SD | 70.5 ± 2.4 | 70.5 ± 2.3 | 70.9 ± 2.5 | 70.8 ± 2.4 | 69.5 ± 3.2 | 69.6 ± 3.05 | 69.6 ± 1.9 | 69.6 ± 1.8 |
| | N | 11 | 2,412 | 9 | 1,965 | 8 | 1,725 | 6 | 1,290 |
| | Relative humidity [%] | | | | | | | | |
| | Mean ± SD | 37.6 ± 7.7 | 37.6 ± 7.3 | 37.5 ± 7.8 | 37.5 ± 7.5 | 36.5 ± 6.0 | 36.4 ± 5.6 | 31.6 ± 4.9 | 31.5 ± 4.6 |
| | N | 11 | 2,412 | 9 | 1,965 | 8 | 1,725 | 8 | 1,290 |

Note: All data pertain to school hours (8:30am to 3:45pm). Lower sample is shown for temperature, humidity, and PM1.0 data (compared to PM2.5 data) data due to missingness. Differences between half-day and minute-level averages are due to rounding errors.

null for $PM_{1.0}$, indicating no additional decreases in either size fraction when both appliances were used simultaneously. Results were generally similar when adjusting for relative humidity.

## Discussion

This was one of the first studies to investigate the effects of different low-cost indoor air quality interventions on $PM_{2.5}$ and $PM_{1.0}$ concentrations in a classroom in a non-urban elementary school in the Northeastern United States. The mean half-day concentrations ranged from 3.4–4.1 $\mu g/m^3$ for $PM_{2.5}$ and 3.4–3.9 $\mu g/m^3$ for $PM_{1.0}$. On average, addition of the fan (with the air cleaner on low) decreased $PM_{2.5}$ by 0.53 $\mu g/m^3$ [95% CI: -0.64, -0.42] and use of the filter on high (compared to low) alone decreased $PM_{2.5}$ by 0.10 $\mu g/m^3$ [95% CI: -0.20, 0.005]. For $PM_{1.0}$, addition of the fan (with the air cleaner on low) decreased concentrations by 0.18 $\mu g/m^3$ [95% CI: -0.36, -0.01] and use of the filter on high (compared to low) alone decreased

**Table 2. One-way unadjusted analysis of variance (ANOVA) comparing four scenarios of HEPA cleaner and air conditioning unit fan use in elementary school classrooms in Rhode Island, USA.**

| Size fraction | Degrees of freedom | F value | Pr (f > F) |
|---|---|---|---|
| PM2.5 | 3 | 0.95 | 0.43 |
| PM1.0 | 3 | 0.45 | 0.72 |

Note: ANOVAs include comparisons of averaged concentrations (over half-days) from each of the four trials (air cleaner on high, fan on; air cleaner on low, fan on; air cleaner on high, fan off; air cleaner on low, fan on) in two classrooms.

**Table 3. Trial results from different scenarios of HEPA cleaner and air conditioning unit fan use in elementary school classrooms in Rhode Island, USA.**

| | Particulate matter size fraction [ug/m³] | | | |
|---|---|---|---|---|
| | PM2.5 | PM1.0 | PM2.5 | PM1.0 |
| | (1) | (2) | (3) | (4) |
| Fan on | -0.53*** | -0.18** | -0.91*** | -0.57*** |
| | (-0.64, -0.42) | (-0.36, -0.01) | (-1.03, -0.78) | (-0.77, -0.38) |
| Filter high | -0.10* | -0.38*** | 0.05 | -0.35*** |
| | (-0.20, 0.005) | (-0.55, -0.21) | (-0.07, 0.16) | (-0.54, -0.15) |
| Fan on * filter high | 0.39*** | 0.04 | 0.26*** | 0.17 |
| | (0.24, 0.54) | (-0.21, 0.28) | (0.08, 0.45) | (-0.13, 0.47) |
| Classroom 2 | -1.03*** | 1.77*** | -1.13*** | 1.77*** |
| | (-1.11, -0.95) | (1.64, 1.89) | (-1.23, -1.04) | (1.62, 1.93) |
| Relative humidity | | | -0.02*** | 0.04*** |
| | | | (-0.03, -0.02) | (0.03, 0.05) |
| Constant | 4.43*** | 3.16*** | 5.38*** | 1.73*** |
| | (4.17, 4.70) | (2.85, 3.46) | (5.06, 5.70) | (1.24, 2.23) |
| Observations | 9,970 | 9,719 | 7,392 | 7,143 |

*p < 0.1;

**p<0.05;

***p<0.01

Note: Results are from linear mixed models including a random intercept for half-day (morning or afternoon). Model results shown include one-minute level data with AR1 correlation structure. Differences in the number of observations are due to missing relative humidity data (column 3 and 4).

concentrations by 0.38 μg/m³ [95% CI: -0.55, -0.21]. We observed that the lowest classroom $PM_{2.5}$ and $PM_{1.0}$ concentrations occurred when both the fan and portable HEPA air cleaner were used simultaneously. However, for $PM_{2.5}$, the lowest concentrations were seen when the fan was on and the air cleaner was on low because most of the benefit was through the use of the fan. Overall, the trends observed in this study give insight into non-urban classroom air quality and how different commercially available, cost-effective IAQ interventions could be incorporated into classrooms to effectively reduce PM concentrations.

Our results were consistent with previous studies on the effects of ventilation systems and portable air cleaners on indoor $PM_{2.5}$ and $PM_{1.0}$ levels in various settings. However, it is important to note that these studies did not evaluate fan use and occurred in areas with higher ambient air pollution. The application of a new mechanical ventilation system with a fine F8 (MERV 14) filter reduced classroom $PM_{2.5}$ concentrations by 30% compared to baseline in an elementary school in Amsterdam with average background $PM_{2.5}$ ranging from 11.2 to 17.9 μg/m3 [25]. Barkjohn et al. 2020 also demonstrated that the use of HEPA fitted portable air cleaners in homes in Beijing, where outdoor $PM_{2.5}$ ranged from 25 to >200 μg/m3, significantly reduced indoor/outdoor $PM_{2.5}$ ratios by 72% when windows were closed [26]. In a crossover study within two residential homes, Hart et al. 2011 demonstrated that a portable air cleaner can be efficient in reducing particulate matter concentrations by >60% for $PM_{2.5}$ and $PM_{1.0}$ in homes that used wood stoves as a primary heating source [27]. A pilot study in four kindergarten classrooms in Poland found that air quality in classrooms with a HEPA air cleaner was on average almost 40–50% better than in those without any procedures to decrease air pollutant concentrations [19]. These studies support our findings that use of air cleaners indoors in a classroom reduces $PM_{2.5}$ and $PM_{1.0}$ concentrations, although we observed smaller decreases of approximately 13% and 2% in $PM_{2.5}$ concentrations when the fan was on and the

air cleaner was on high, respectively. In addition, they do not provide an explanation for why the greatest reduction in $PM_{2.5}$ concentrations occurred with the low HEPA air cleaner setting and not the higher setting, which filters a greater volume of air per hour.

We speculate that using a lower CADR setting could provide a greater reduction in $PM_{2.5}$ concentrations in classrooms with low background $PM_{2.5}$ due to particle removal efficiency. A lab study demonstrated that increased media velocity was associated with a decrease in HEPA filter efficiency from 99.999%, although the efficiency rate did not decrease below 99.977% [28]. A higher CADR setting increases the volume of air filtered during a given period, resulting in greater air exchange. However, as air is pulled through the HEPA filter more quickly, the filter may be less effective at removing particles, especially at low concentrations. A setting that reduces the rate of air filtration would theoretically move the air through the filter more slowly, allowing for more effective removal of particles. One study found that using portable HEPA air cleaners at home on "Auto" mode significantly reduced indoor $PM_{2.5}$ concentrations compared to homes that manually selected the air cleaner settings [29]. The authors concluded that auto-mode filtration provided the most efficient results, especially when indoor sources (e.g., cooking) were present. While the air cleaners used in this study did not have an "Auto" mode, it is possible the lower CADR setting was more efficient for $PM_{2.5}$ given the low background concentrations and possible settling on indoor surfaces. However, we were not able to address the influence of classroom occupancy and activity on indoor PM levels. It is possible the higher CADR setting may have been more efficient at removing $PM_{1.0}$ since smaller particles are more likely remain suspended. Croxford et al. reported that use of an electrostatic air cleaner in an office setting in London reduced smaller particles ($PM_{2.0}$) more effectively than larger particles ($PM_{2\text{-}10}$) [30]. Similarly, Park et al. 2020's study of 34 elementary schools in Korea found that air cleaners appeared to remove $PM_{2.5}$ more effectively than $PM_{10}$ [12]. Meanwhile, a crossover study of five elementary schools found that the use of electrostatic air cleaners in school classrooms substantially decreased airborne indoor PM concentrations but not indoor surface dust levels [31]. Our findings might imply that, at low concentrations, larger particles are removed more effectively with a lower CADR setting while smaller particles are removed more effectively with a higher CADR setting since they are less likely to settle on indoor surfaces.

This study also evaluated A/C fan ventilation; however, research is limited on the effect of fan use on indoor classroom PM concentrations. A study of five classrooms in Hong Kong, a region with high ambient concentrations of PM, suggested that even in the presence of ventilation (air conditioning or ceiling fans) indoor $PM_{10}$ concentrations mirrored outdoor $PM_{10}$ concentrations [32]. If we expect that classroom $PM_{2.5}$ concentrations may mirror outdoor $PM_{2.5}$ concentrations in areas with high pollution levels, this may also be the case in a city with very low ambient $PM_{2.5}$ concentrations. If a location has low background $PM_{2.5}$ concentrations, ambient PM from natural ventilation may dilute indoor PM concentrations. During the study period, there were four days with average daily ambient $PM_{2.5}$ concentrations that were lower than the average daily classroom concentrations (see Fig 1). Therefore, natural ventilation with ambient air may have contributed to lower classroom PM concentrations. The presence of a window A/C unit could have also introduced small quantities of outdoor air into the classroom if not sealed properly, or through leaks in the A/C unit. If ambient air that had lower $PM_{2.5}$ concentrations was introduced into the classroom, the classroom $PM_{2.5}$ concentrations could be diluted, resulting in lower indoor PM2.5 concentrations.

We speculate the dual effect of the air cleaner running on "low" and the fan "on" may result from low background PM concentrations compounded by the combined filtration and air circulation from both the air cleaner and the A/C unit. The fans were located approximately one meter above ground level and therefore could have introduced diluted air into the classroom

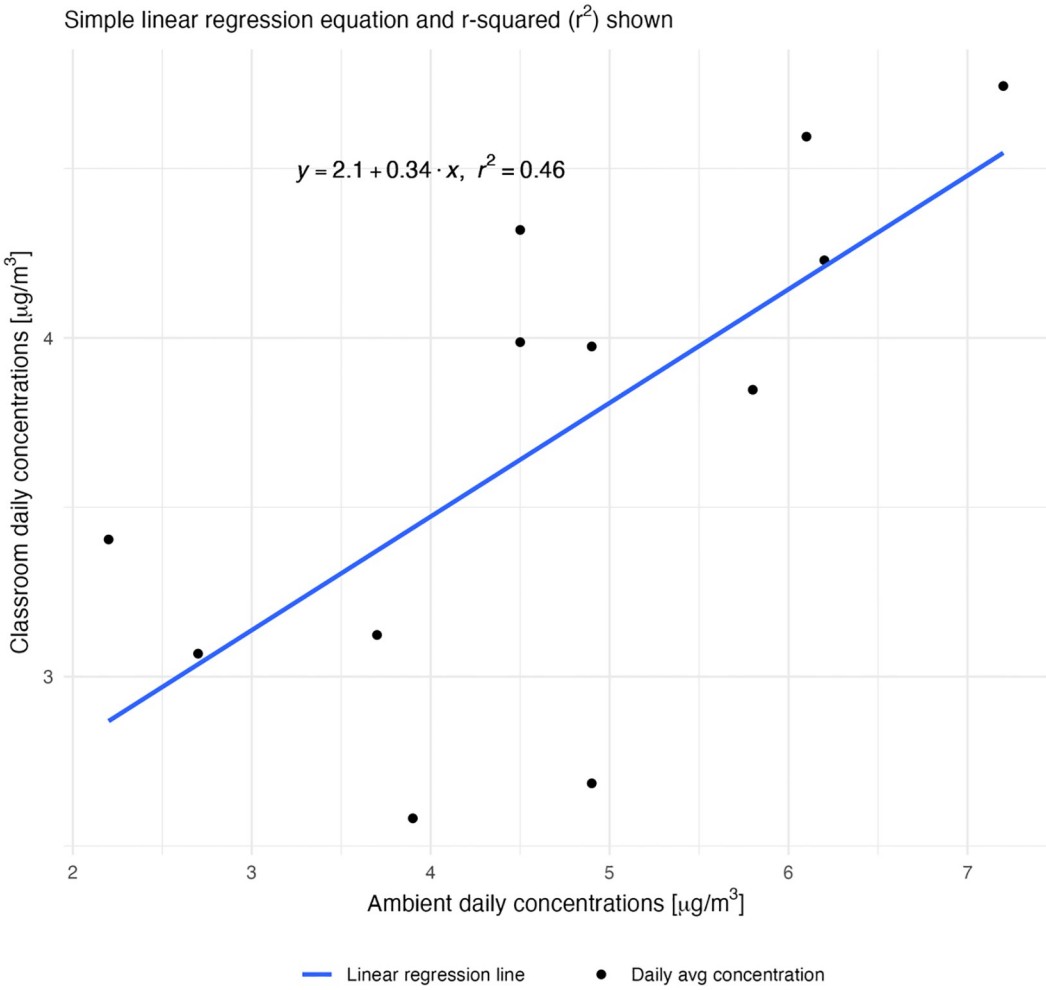

**Fig 1. Ambient vs. classroom daily average PM$_{2.5}$ concentrations.**

without disturbing settled particles on the ground. Background PM concentrations in the classrooms may have further influenced the effectiveness of the air cleaners. As detailed above, low concentrations of PM may not be effectively filtered by higher CADR settings due to increased flow rate and potential settling of coarser particles. We were not able to evaluate flow dynamics between the A/C fan, the air cleaner, and the mechanical ventilation systems in each classroom, so it is difficult to say whether the additional air circulation provided by the A/C fan contributed to resuspension of such particles. Additionally, the filter in the A/C unit, even if crude, could have removed some larger particles. Further, the concentrations observed in this study could relate to the condition of the HEPA filters. The air cleaner machines were purchased for the classrooms as a response to the COVID-19 pandemic, however, it is unknown when (or whether) the HEPA filters in the air cleaners were last cleaned or changed. If the filters were clogged with debris, filtration efficiency may have been affected.

Our study was strengthened by use of the same classrooms over a two-week period in the same elementary school. The classrooms were of approximately equal size (area: 140 m$^2$ and 152 m$^2$; volume: 427 m$^3$ and 463 m$^3$) and layout, and there was a consistent number of students (14–16) in each classroom over the sampling period (S1 Fig). This limited potential

confounding in our data by changes in classroom size and occupancy. The use of a randomized schedule further limited potential sources of unmeasured confounding. Another strength of this study is that we evaluated two fractions of particulate matter to determine whether air quality interventions differed by PM particle size.

In contrast, a major limitation of this study was our inability to evaluate the classrooms under the condition of air cleaner "off" due to COVID-19 and health guidelines, eliminating the possibility to compare the effects of HEPA filter settings to a baseline without the air cleaner. We also did not evaluate the overall school filtration system, which could have influenced PM concentrations in the classrooms and could have attenuated the impacts of the air quality interventions seen in this study. Correction factors calculated from baseline data were applied to two of the aerosol monitors to adjust for differences between the units in measuring PM concentrations, which could have influenced the results.

Furthermore, given the nature of the COVID-19 pandemic and visitor restrictions in place, we were unable to be present in the classrooms during the study period and were not able to gather real-time CADR values or evaluate the sound level of the air purifier and A/C unit at different levels. Sound at different levels was also not evaluated in this study as the primary purpose was to evaluate PM concentrations. Additionally, the teachers were required to operate air purifiers while the classroom was occupied to reduce risk of COVID-19 exposure, however, teachers had the ability to run the purifiers at a variety of settings and we selected only a few settings to study as part of our intervention. It should also be noted that we did not receive noise complaints from the teachers who volunteered their classrooms for the study and the sound was not loud enough to interfere with speaking or listening in the classroom. Even though the use of the air purifier and fan introduces additional noise into the classroom, the devices improve indoor air quality and health. Lastly, despite the classrooms being of approximately equal length, width, and height, the air cleaners in the two classrooms were positioned at different heights (one on the floor and another elevated approximately 0.5 m). However, we were able to include a fixed effect to correct for baseline classroom differences in our mixed model analyses.

Despite the limitations, this study is impactful because it adds to the limited body of literature surrounding $PM_{2.5}$ and $PM_{1.0}$ concentrations in classroom settings in the United States. This study provides a foundation for future research in the field, especially since the Biden administration initiated a new focus on indoor air quality with an emphasis on schools. Though we did not consider the effects of seasonality given the small sampling period, seasonality should be considered in future studies. Classroom PM concentrations have been shown to vary by season, with higher concentrations of PM occurring in the winter compared to the summer [33]. Occupant behavior (e.g., window opening, time/activity spent indoors) would also vary significantly by season. It would also be worthy to more closely evaluate the impacts of occupancy—the number of teachers and students per classroom—on PM concentrations. Multiple studies have shown that mechanical forces are a primary driver of classroom air quality [34]. Although the teachers were asked to track times of events that may impact PM concentrations (e.g., bus arrivals, times during which all students typically leave the classroom), the data was not sufficient to make any conclusions about the influence of activity on PM concentrations in our study. Therefore, it is important for future studies to look at how the number of occupants per classroom, and student activity, influences PM concentrations and the effectiveness of different air quality interventions.

Our study only involved two classrooms because of availability of equipment and could have been strengthened if more classrooms were evaluated. As a result, the findings from two classrooms in one school limit the generalizability of the study findings even though we included multiple day measurements. Furthermore, to better evaluate the effectiveness of air

quality intervention, air cleaner devices should be placed in the same location and height in each classroom to ensure consistency in sampling. Monitoring the CADR levels at different operating levels would help contextualize the impact of using the air cleaners at different settings. It is also important to note that this study was conducted in a non-urban area with low ambient PM concentrations and is not reflective of how these air quality interventions may work in urban settings or areas with higher ambient PM concentrations.

## Conclusion

This study demonstrated that the lowest average classroom PM concentrations occurred when the A/C fan was on in the classrooms. The highest concentrations of both $PM_{2.5}$ and $PM_{1.0}$ concentrations occurred when the filter was used on low without concurrent use of the fan. Therefore, the use of a fan in a non-urban classroom with low background PM concentrations may help to reduce overall classroom PM concentrations. Specifically, within the context of this study for a school with a low background $PM_{2.5}$ concentration, the use of an air cleaner on the "low" setting instead of a "high setting" could be used to supplement classroom ventilation and filtration while also lowering operating costs. In settings where ambient air quality is already at an acceptable level, using air cleaners at a higher setting may not be as efficient and may not significantly improve air quality compared to a lower setting.

## Supporting information

**S1 Fig. Schematic of classroom A and classroom B.**
(TIF)

## Acknowledgments

The authors would like to thank Sara Gillooly and Jordana Leader for their technical and study design support throughout the duration of this project. Additional support was appreciated from the Superintendent, School Principal, teachers, and staff at the elementary school in RI who made this study possible.

## Author Contributions

**Conceptualization:** Alexandra Azevedo, Jahred Liddie, Jason Liu, Jessica E. Schiff, Jaime E. Hart.

**Data curation:** Alexandra Azevedo, Jahred Liddie, Jason Liu, Jessica E. Schiff.

**Formal analysis:** Jahred Liddie, Jason Liu, Gary Adamkiewicz, Jaime E. Hart.

**Funding acquisition:** Jahred Liddie.

**Investigation:** Alexandra Azevedo, Jahred Liddie, Jason Liu, Jessica E. Schiff.

**Methodology:** Alexandra Azevedo, Jahred Liddie, Jason Liu, Jessica E. Schiff, Gary Adamkiewicz, Jaime E. Hart.

**Resources:** Alexandra Azevedo, Gary Adamkiewicz, Jaime E. Hart.

**Software:** Jahred Liddie, Jason Liu, Jaime E. Hart.

**Supervision:** Gary Adamkiewicz, Jaime E. Hart.

**Validation:** Gary Adamkiewicz, Jaime E. Hart.

**Visualization:** Alexandra Azevedo, Jahred Liddie, Jason Liu, Jessica E. Schiff, Gary Adamkie-wicz, Jaime E. Hart.

**Writing – original draft:** Alexandra Azevedo, Jahred Liddie, Jason Liu, Jessica E. Schiff.

**Writing – review & editing:** Alexandra Azevedo, Jahred Liddie, Jason Liu, Jessica E. Schiff, Gary Adamkiewicz, Jaime E. Hart.

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
