## [Decision Letter · Decision Letter 0]

4 May 2022

PONE-D-22-05922Effects of portable air cleaners and A/C unit fans on classroom concentrations of particulate matter in a non-urban elementary schoolPLOS ONE

Dear Dr. Schiff,

Thank you for submitting your manuscript to PLOS ONE. After careful consideration, we feel that it has merit but does not fully meet PLOS ONE’s publication criteria as it currently stands. Therefore, we invite you to submit a revised version of the manuscript that addresses the points raised during the review process.

We look forward to receiving your revised manuscript.

Kind regards,

MARIA LUISA ASTOLFI, Ph.D.

Academic Editor

PLOS ONE

Journal Requirements:

2. You indicated that ethical approval was not necessary for your study. We understand that the framework for ethical oversight requirements for studies of this type may differ depending on the setting and we would appreciate some further clarification regarding your research. Could you please provide further details on why your study is exempt from the need for approval and confirmation from your institutional review board or research ethics committee (e.g., in the form of a letter or email correspondence) that ethics review was not necessary for this study? Please include a copy of the correspondence as an ""Other"" file.

Furthermore we recommend that the consent statement is revised to indicate that the head teachers of the school provided in parentis loco consent to conduct this study.

“Jahred Liddie, a co-author, was supported by a Training Grant in Environmental Epidemiology (T32 E007069) from the National Institute of Environmental Health Sciences. No other specific funding was received for this work by any of the co-authors. Funds from the Department of Environmental Health, Harvard TH Chan School of Public Health and the Harvard Chan NIEHS Center (NIH/NIEHS P30 ES000002) were used to purchase and maintain the equipment used for the study.”

“Jahred Liddie was also supported by a Training Grant in Environmental Epidemiology (T32 E007069) from the National Institute of Environmental Health Sciences. The authors would like to thank Jaime E. Hart and Gary Adamkiewicz for their contributions of expertise and support in this project”

“Jahred Liddie, a co-author, was supported by a Training Grant in Environmental Epidemiology (T32 E007069) from the National Institute of Environmental Health Sciences. No other specific funding was received for this work by any of the co-authors. Funds from the Department of Environmental Health, Harvard TH Chan School of Public Health and the Harvard Chan NIEHS Center (NIH/NIEHS P30 ES000002) were used to purchase and maintain the equipment used for the study.”

Reviewers' comments:

Reviewer's Responses to Questions

**Comments to the Author**

1. Is the manuscript technically sound, and do the data support the conclusions?

Reviewer #1: Partly

Reviewer #2: Yes

Reviewer #3: Partly

2. Has the statistical analysis been performed appropriately and rigorously? 

Reviewer #1: Yes

Reviewer #2: Yes

Reviewer #3: Yes

3. Have the authors made all data underlying the findings in their manuscript fully available?

Reviewer #1: Yes

Reviewer #2: Yes

Reviewer #3: No

4. Is the manuscript presented in an intelligible fashion and written in standard English?

Reviewer #1: Yes

Reviewer #2: Yes

Reviewer #3: Yes

5. Review Comments to the Author

Reviewer #1: Abstract:

The structure of the abstract must be changed. Start with the problem you have found that triggers your research and finalize with the objective, methodology and conclusions.

This is the second sentence of the abstract: “Our objective was to determine if use of air cleaners with HEPA filters and air conditioning (A/C) units were associated with changes in particulate matter (PM) air pollution concentrations in a real-world environment.”

Could you change the order and include the sentence as the main goal of the article? What is the method to achieve that goal?

Introduction:

Lines 101-103: “To the best of our knowledge, this is the first study to provide data on the effectiveness of portable air cleaners and fans operating in tandem to reduce PM2.5 and PM1.0 concentrations in occupied classrooms.” This could be the main objective of the article. I suggest not to mention “To the best of our knowledge, this is the first study…” Just state the objective of the article.

Please, unify the format of the tables.

Figure 1. Please explain the equation and the parameter “r”.

Conclusions:

Use some of the numerical results to reinforce the statements.

Include some limitations you might have found in the methodology and future improvements.

All the data stated in the abstract should have comments in the conclusion section:

“The mean half-day concentrations ranged from 3.4 - 4.1 µg/m 2 for PM 2.5 and 3.4 - 3.9 µg/m 2 for PM 1.0 On average, use of the fan decreased PM 2.5 by 0.53 µg/m 3 [95% CI: -0.64, -0.42] and use of the filter on high (compared to low) decreased PM 2.5 by 0.10 µg/m 3 [95% CI: -0.20, 0.005]. For PM 1.0 , use of the fan decreased concentrations by 0.18 µg/m 3 [95% CI: -0.36, -0.01] and use of the filter on high (compared to low) decreased concentrations by 0.38 µg/m 3 [95% CI: -0.55, -0.21].”

Reviewer #2: The manuscript titled ‘Effects of portable air cleaners and A/C unit fans on classroom concentrations of particulate matter in a non-urban elementary school’ documents the effects of air cleaner in the low/high mode and A/C fan in the on/off mode using linear mixed regression models. Interestingly, the manuscript does not discuss potential impacts of students’ activity on classroom PM and on the effects of air cleaner. Because student activity is an important factor affecting classroom PM concentrations and effect of air cleaners on classroom PM, this needs to be discussed in Discussion. The manuscript may also consider the following comments to improve clarity and flow of the manuscript.

Major comments:

1. Lines 44-47: It should be mentioned that the effect of fan was adjusted for air cleaner; likewise, the effect of air cleaner was adjusted for A/C fan.

2. Lines 48-51: General statement (lines 48-49) of the study findings and the conclusion (lines 50-51) don’t seem to agree in the current writing. It looks like that the conclusion should be modified to more specific one because the effects of concurrent use of an A/C fan and air cleaner on reducing the PM were dependent of the mode of the air cleaner and PM size. The proper conclusion would have implication in saving energy.

3. Lines 48-49: Does the ‘additional decreases in PM concentrations’ mean no interaction effect between fan on and air cleaner on high? But the interaction was only significant for PM2.5 but not for PM1.0. If this statement was from the interaction model outputs, it should be also specific to PM size.

4. Lines 79-84: The sentences are about health effects of PM, which has nothing to do with use of air cleaner or improved ventilation that seemed to be the theme for the previous sentences within the paragraph. Thus, it would flow better if these were moved to the next paragraph (after the last sentence of the next paragraph that is describing PM health effect).

5. Wondering if the authors examined interaction between classroom humidity and fan or between humidity and air cleaner in the regression models?

6. Lines 198-205: It seemed that fan effect was bigger for PM2.5 then PM1.0, but air cleaner effect was bigger for PM1.0 than PM2.5. Isn’t this worth to discuss further in Discussion?

7. The title of Table 2 should state ‘Unadjusted ANOVA’ if the ANOVA models were not adjusted for anything.

8. Lines 235-243: The studies discussed are all home studies. Are there any studies of evaluating the effect of air cleaners in classrooms? Unless discussion is strictly limited to HEPA air cleaners, there are some studies examining effects of air cleaners in classrooms (e.g., Wargocki et al., HVAC R Res. 14 (2008): 327-344; Mattsson and Hygge, 2005 Indoor Air Conference Proceedings, pp 1111-1115; Park et al., Building and Environment 167, 2020: 106437), which should be discussed.

9. Lines 256-257: The discussion doesn’t agree with the current study finding. It was reported in Results that the air cleaner on high mode reduced PM1.0 concentrations more than PM2.5. Thus, your argument in this sentence about ‘less effective at removing particles for smaller size PM by air cleaner on high mode’ is not supported by your own finding.

10. Line 268: ‘….some average daily ambient …’. Specify ‘some’ in this sentence because there are only two days when ambient PM2.5 concentrations were lower than classroom PM2.5.

11. Lines 261-273: This paragraph was not clear. What is the main discussion point of the paragraph? The paragraph may need to be modified for clarification.

12. In the limitation section, the small number of classrooms in the study also needs to be mentioned as a limitation. Findings from only two classrooms in one school may not be generalizable although the study has multiple day measurements.

Reviewer #3: Dear Author,

Your manuscript follows a very interesting approach. However, important information is missing to be able to assess the results and evaluate them for a school. For example, a room sketch is missing, as well as information about the ceiling height, with the coordinates for the air purifier, A/C unit and also the measuring points. Furthermore, you do not address the CADR values that are realised at different levels of the air cleaner. The sound level at the different levels is also not mentioned, although this is a critical factor for use in classrooms. The same applies to the A/C unit. I read online that a sound pressure level of 57 dB(A) is generated at the "Low" level, which is clearly too high and would in turn have an impact on the students' ability to concentrate. In my view, when you address the specific case in schools, you have to take these points into account.

From my point of view, a major revision is necessary so that a general gain in knowledge emerges from the manuscript.

6. PLOS authors have the option to publish the peer review history of their article (what does this mean?). If published, this will include your full peer review and any attached files.

Reviewer #1: No

Reviewer #2: No

Reviewer #3: No

---

## [Author Response · Author response to Decision Letter 0]

10 Aug 2022

Response to Editors’ Comments

2. You indicated that ethical approval was not necessary for your study. We understand that the framework for ethical oversight requirements for studies of this type may differ depending on the setting and we would appreciate some further clarification regarding your research. Could you please provide further details on why your study is exempt from the need for approval and confirmation from your institutional review board or research ethics committee (e.g., in the form of a letter or email correspondence) that ethics review was not necessary for this study? Please include a copy of the correspondence as an ""Other"" file.

Furthermore, we recommend that the consent statement is revised to indicate that the head teachers at the school provided in parentis loco consent to conduct this study.

Response: Please see the PDF file “IRB Exemption”, categorized as “other”

“Jahred Liddie, a co-author, was supported by a Training Grant in Environmental Epidemiology (T32 E007069) from the National Institute of Environmental Health Sciences. No other specific funding was received for this work by any of the co-authors. Funds from the Department of Environmental Health, Harvard TH Chan School of Public Health and the Harvard Chan NIEHS Center (NIH/NIEHS P30 ES000002) were used to purchase and maintain the equipment used for the study.”

Response: Thank you for this comment. We have added the provided statement to the funding section of the paper and included a Role of Funder statement in the cover letter.

“Jahred Liddie was also supported by a Training Grant in Environmental Epidemiology (T32 E007069) from the National Institute of Environmental Health Sciences. The authors would like to thank Jaime E. Hart and Gary Adamkiewicz for their contributions of expertise and support in this project”

“Jahred Liddie, a co-author, was supported by a Training Grant in Environmental Epidemiology (T32 E007069) from the National Institute of Environmental Health Sciences. No other specific funding was received for this work by any of the co-authors. Funds from the Department of Environmental Health, Harvard TH Chan School of Public Health and the Harvard Chan NIEHS Center (NIH/NIEHS P30 ES000002) were used to purchase and maintain the equipment used for the study.”

Response: These statements were moved to the funding section. 

Response: All the data necessary for the analyses have been uploaded. There are three datasets. 

Comments to the Author

Reviewer #1: Abstract:

The structure of the abstract must be changed. Start with the problem you have found that triggers your research and finalize with the objective, methodology and conclusions.

This is the second sentence of the abstract: “Our objective was to determine if use of air cleaners with HEPA filters and air conditioning (A/C) units were associated with changes in particulate matter (PM) air pollution concentrations in a real-world environment.”

Could you change the order and include the sentence as the main goal of the article? What is the method to achieve that goal?

Response: 

Abstract (Reorganized) 

Given the increased useing of air cleaners in classrooms during the COVID-19 pandemic as a prevention measure, this study aimed to investigate the the effects of portable air cleaners with HEPA filters and window A/C fans on real-time (1 minute) concentrations of PM less than 2.5 microns (PM2.5) or less than 1 microns (PM1.0) in two classrooms in a non-urban elementary school in Rhode Island. For half of each school day, settings were randomized to “high” or “low” for the air cleaner and “on” or “off” for the fan. Descriptive statistics and linear mixed models were used to evaluate the impacts of each set of conditions on PM2.5 and PM1.0 concentrations. The mean half-day concentrations ranged from 3.4 - 4.1 µg/m3 for PM2.5 and 3.4 - 3.9 µg/m3 for PM1.0 On average, use of the fan alone decreased PM2.5 by 0.53 µg/m3 [95% CI: -0.64, -0.42] and use of the filter on high (compared to low) alone decreased PM2.5 by 0.10 µg/m3 [95% CI: -0.20, 0.005]. For PM1.0, use of the fan alone decreased concentrations by 0.18 µg/m3 [95% CI: -0.36, -0.01] and use of the filter on high (compared to low) alone decreased concentrations by 0.38 µg/m3 [95% CI: -0.55, -0.21]. In general, simultaneous use of the fan and filter on high did not result in additional decreases in PM concentrations compared to the simple addition of each appliance’s individual effect estimates. use of either appliance individually. Our study suggests that concurrent or separate use of an A/C fan and air cleaner in non-urban classrooms with low background PM may reduce classroom PM concentrations.

Introduction:

Lines 101-103: “To the best of our knowledge, this is the first study to provide data on the effectiveness of portable air cleaners and fans operating in tandem to reduce PM2.5 and PM1.0 concentrations in occupied classrooms.” This could be the main objective of the article. I suggest not to mention “To the best of our knowledge, this is the first study…” Just state the objective of the article.

Response: The objective of this study is to provide data on the effectiveness of portable air cleaners and fans operating in tandem to reduce PM2.5 and PM1.0 concentrations in occupied classrooms.

Please, unify the format of the tables.

Response: We have adjusted the formatting and file types of the tables to be unified.

Figure 1. Please explain the equation and the parameter “r”.

Response: We have added a caption within the figure describing both these components.

Conclusions:

Use some of the numerical results to reinforce the statements.

Response: Please see the tracked changes document that demonstrate the use of numerical results to reinforce the statements. 

Include some limitations you might have found in the methodology and future improvements.

Response (bold are the changes): This study adds to the limited body of literature surrounding PM2.5 and PM1.0 concentrations in classroom settings in the United States and provides a foundation for future research in the field. Though we did not consider the effects of seasonality given the small sampling period, seasonality should be considered in future studies. Classroom PM concentrations have been shown to vary by season, with higher concentrations of PM occurring in the winter compared to the summer [32]. Occupant behavior (e.g., window opening) would also vary significantly by season. It would also be worthy to more closely evaluate the impacts of occupancy—the number of teachers and students per classroom—on PM concentrations. Multiple studies have shown that mechanical forces are a primary driver of classroom air quality, therefore it is important to look at how the number of students per classroom influences PM concentrations and the effectiveness of different air quality interventions [32]. This study only involved two classrooms as a result of availability of equipment and could have been strengthened if more classrooms were evaluated. As a result, the findings from two classrooms in one school limit the generalizability of the study findings even though the study has multiple day measurements.. Furthermore, to better evaluate the effectiveness of air quality intervention, the air quality device should be placed in the same location and at the height in each classroom to ensure consistency in sampling method. Monitoring the CADR levels at different air quality device operating levels would help contextualize the impact of using the air quality devices at different settings. It is also important to note that this study was conducted in a non-urban area with low ambient PM concentrations and is not reflective of how these air quality interventions may work in urban settings or areas with higher ambient PM concentrations. 

All the data stated in the abstract should have comments in the conclusion section:

“The mean half-day concentrations ranged from 3.4 - 4.1 µg/m 2 for PM 2.5 and 3.4 - 3.9 µg/m 2 for PM 1.0 On average, use of the fan decreased PM 2.5 by 0.53 µg/m 3 [95% CI: -0.64, -0.42] and use of the filter on high (compared to low) decreased PM 2.5 by 0.10 µg/m 3 [95% CI: -0.20, 0.005]. For PM 1.0 , use of the fan decreased concentrations by 0.18 µg/m 3 [95% CI: -0.36, -0.01] and use of the filter on high (compared to low) decreased concentrations by 0.38 µg/m 3 [95% CI: -0.55, -0.21].”

Response: Thank you - we have added this to the conclusions section.

Reviewer #2: 

The manuscript titled ‘Effects of portable air cleaners and A/C unit fans on classroom concentrations of particulate matter in a non-urban elementary school’ documents the effects of air cleaner in the low/high mode and A/C fan in the on/off mode using linear mixed regression models. Interestingly, the manuscript does not discuss potential impacts of students’ activity on classroom PM and on the effects of air cleaner. Because student activity is an important factor affecting classroom PM concentrations and effect of air cleaners on classroom PM, this needs to be discussed in Discussion. The manuscript may also consider the following comments to improve clarity and flow of the manuscript.

Response: Thank you – we have added not being able to record student activity (due to COVID-19 limitations at the time) as a limitation to the study, which can be found in the Discussion. It should be noted that we asked the teachers to track times of events that may impact PM concentrations (e.g. bus arrivals, times during which all students typically leave the classroom), but the data was not sufficient to make any conclusions.

Major comments:

1. Lines 44-47: It should be mentioned that the effect of fan was adjusted for air cleaner; likewise, the effect of air cleaner was adjusted for A/C fan.

Response: Thank you – please see lines 40-46 in the tracked changes document to see the clarification on the adjustments for air clear and the A/C fan. 

2. Lines 48-51: General statement (lines 48-49) of the study findings and the conclusion (lines 50-51) don’t seem to agree in the current writing. It looks like that the conclusion should be modified to more specific one because the effects of concurrent use of an A/C fan and air cleaner on reducing the PM were dependent of the mode of the air cleaner and PM size. The proper conclusion would have implication in saving energy.

Response: Simultaneous use of the fan and air cleaner did not result in significant, additional decreases in PM2.5 - as stated in the results section, the interaction term was actually significantly positive. Regarding PM1.0, no significant differences associated with simultaneous use of the air cleaner and fan were found. We have updated the phrasing of the results in the Abstract as well as in the Results section to better reflect this finding.

3. Lines 48-49: Does the ‘additional decreases in PM concentrations’ mean no interaction effect between fan on and air cleaner on high? But the interaction was only significant for PM2.5 but not for PM1.0. If this statement was from the interaction model outputs, it should be also specific to PM size.

Response: Thank you for this comment. As clarification and as described in the Result section, the interaction term was significantly for PM2.5 and null for PM1.0. This indicated that concurrent use of the appliances, in our study setting, did not result in significantly lower PM concentrations for either size fraction below the simple addition of each separate effect estimates. In line with previous comment, we have rephrased the results description in the Abstract and Result sections to reflect this more clearly.

4. Lines 79-84: The sentences are about health effects of PM, which has nothing to do with use of air cleaner or improved ventilation that seemed to be the theme for the previous sentences within the paragraph. Thus, it would flow better if these were moved to the next paragraph (after the last sentence of the next paragraph that is describing PM health effect).

Response: Given the challenges of installing new HVAC systems or filters in school buildings, portable air cleaners and fans have become popular, less expensive alternatives to reduce PM concentrations. Air cleaners with high-efficiency particulate air (HEPA) filters, which are at least 99.97% efficient at capturing particles 0.3 µm and larger in size, offer the possibility of substantially reducing SARS-CoV-2 viral particles as well as improving overall IAQ [7,5]. Beyond the current context of the COVID-19 pandemic, improved air ventilation is associated with lower school absenteeism, better performance on cognitive function tests, and fewer respiratory symptoms, such as those related to asthma, lung inflammation, and allergies [9, 10, 11]. PM2.5 exposure in children has been associated with asthma incidence (OR=1.10, 95% CI:1.01, 1.20), prevalence of asthma symptoms (OR=1.08, 95% CI:1.02, 1.16), and rhinitis (OR=1.15, 95% CI: 1.05,1.26) [12]. A review study of 33 articles further provided evidence that exposure to particulate air pollution has adverse impacts on children’s respiratory health, with stronger negative effects seen among children with asthma [13]. 

PM is one of the most common pollutants that could potentially degrade air quality in classrooms [14]. Indoor PM levels are influenced by several factors, such as ambient air pollution levels, air exchange rates, occupancy, type and intensity of indoor activities, and particle sizes [14, 15, 16]. One study in Munich, Germany found that PM concentrations in classrooms are about six times higher than outdoor concentrations [17]. Children are particularly vulnerable to potential health consequences related to PM exposure due to their immature respiratory and immune systems and greater breathing rates per body weight [11]. Beyond the current context of the COVID-19 pandemic, improved air ventilation is associated with lower school absenteeism, better performance on cognitive function tests, and fewer respiratory symptoms, such as those related to asthma, lung inflammation, and allergies [9, 10, 11]. PM2.5 exposure in children has been associated with asthma incidence (OR=1.10, 95% CI:1.01, 1.20), prevalence of asthma symptoms (OR=1.08, 95% CI:1.02, 1.16), and rhinitis (OR=1.15, 95% CI: 1.05,1.26) [12]. A review study of 33 articles further provided evidence that exposure to particulate air pollution has adverse impacts on children’s respiratory health, with stronger negative effects seen among children with asthma [13]. 

5. Wondering if the authors examined interaction between classroom humidity and fan or between humidity and air cleaner in the regression models?

Response: We tried this and a model with a three-way interaction - generally, the main effects are almost identical, although there are some significant but small interactions with humidity that vary for the size fractions. However, we decided not to include the result due the difficulty to interpret the impact and the concern that if may distract from results. 

6. Lines 198-205: It seemed that fan effect was bigger for PM2.5 then PM1.0, but air cleaner effect was bigger for PM1.0 than PM2.5. Isn’t this worth to discuss further in Discussion?

Response: Thank you - this point has been addressed in the Discussion.

7. The title of Table 2 should state ‘Unadjusted ANOVA’ if the ANOVA models were not adjusted for anything.

Response: Thank you - we have added this distinction.

8. Lines 235-243: The studies discussed are all home studies. Are there any studies of evaluating the effect of air cleaners in classrooms? Unless discussion is strictly limited to HEPA air cleaners, there are some studies examining effects of air cleaners in classrooms (e.g., Wargocki et al., HVAC R Res. 14 (2008): 327-344; Mattsson and Hygge, 2005 Indoor Air Conference Proceedings, pp 1111-1115; Park et al., Building and Environment 167, 2020: 106437), which should be discussed.

Response: The studies discussed were limited to HEPA air cleaners. We discussed one classroom study that also used HEPA air cleaners, but acknowledged a general lack of existing literature. However, we have cited more school studies elsewhere in the Discussion, including a few mentioned by the reviewer.

9. Lines 256-257: The discussion doesn’t agree with the current study finding. It was reported in Results that the air cleaner on high mode reduced PM1.0 concentrations more than PM2.5. Thus, your argument in this sentence about ‘less effective at removing particles for smaller size PM by air cleaner on high mode’ is not supported by your own finding.

Response: Thank you for this comment. We went to this paragraph and revised our language. We had incorrectly discussed particle size instead of particle concentration since our study was in a classroom that had very low PM concentrations. We also added additional context. Please see lines 365- 398 (in track changes document). 

10. Line 268: ‘….some average daily ambient …’. Specify ‘some’ in this sentence because there are only two days when ambient PM2.5 concentrations were lower than classroom PM2.5.

Response: Thank you - we have added this clarification.

11. Lines 261-273: This paragraph was not clear. What is the main discussion point of the paragraph? The paragraph may need to be modified for clarification.

Response: Thank you – The main discussion of the paragraph was to highlight the fact that the study was conducted in a location that has very low ambient PM concentrations and that those low ambient concentrations may have impacted classroom PM concentrations. We edited the paragraph starting at line 404-422 (in the track changes document) to emphasize the low ambient PM concentrations. 

12. In the limitation section, the small number of classrooms in the study also needs to be mentioned as a limitation. Findings from only two classrooms in one school may not be generalizable although the study has multiple day measurements.

Response (bold are the changes): This study adds to the limited body of literature surrounding PM2.5 and PM1.0 concentrations in classroom settings in the United States and provides a foundation for future research in the field. Though we did not consider the effects of seasonality given the small sampling period, seasonality should be considered in future studies. Classroom PM concentrations have been shown to vary by season, with higher concentrations of PM occurring in the winter compared to the summer [32]. Occupant behavior (e.g., window opening) would also vary significantly by season. It would also be worthy to more closely evaluate the impacts of occupancy—the number of teachers and students per classroom—on PM concentrations. Multiple studies have shown that mechanical forces are a primary driver of classroom air quality, therefore it is important to look at how the number of students per classroom influences PM concentrations and the effectiveness of different air quality interventions [32]. This study only involved two classrooms as a result of availability of equipment and could have been strengthened if more classrooms were evaluated. As a result, the findings from two classrooms in one school limit the generalizability of the study findings even though the study has multiple day measurements.. Furthermore, to better evaluate the effectiveness of air quality intervention, the air quality device should be placed in the same location and at the height in each classroom to ensure consistency in sampling method. Monitoring the CADR levels at different air quality device operating levels would help contextualize the impact of using the air quality devices at different settings. It is also important to note that this study was conducted in a non-urban area with low ambient PM concentrations and is not reflective of how these air quality interventions may work in urban settings or areas with higher ambient PM concentrations. 

Reviewer #3: Dear Author,

Your manuscript follows a very interesting approach. However, important information is missing to be able to assess the results and evaluate them for a school. For example, a room sketch is missing, as well as information about the ceiling height, with the coordinates for the air purifier, A/C unit and also the measuring points. Furthermore, you do not address the CADR values that are realized at different levels of the air cleaner. The sound level at the different levels is also not mentioned, although this is a critical factor for use in classrooms. The same applies to the A/C unit. I read online that a sound pressure level of 57 dB(A) is generated at the "Low" level, which is clearly too high and would in turn have an impact on the students' ability to concentrate. In my view, when you address the specific case in schools, you have to take these points into account.

Response: We added sound level as a limitation in our discussion. We were concerned about the impact of the sound in the classroom; however, the air purifiers and A/C fans were not so loud and did not interfere with the ability to teach in the classroom. Furthermore, internet research indicates that 60 dB is conversation volume. We currently do not have a sketch of the classrooms but we included dimensions of the classrooms and location of the air purifiers in the classroom in the methods section.

---

## [Decision Letter · Decision Letter 1]

6 Sep 2022

PONE-D-22-05922R1Effects of portable air cleaners and A/C unit fans on classroom concentrations of particulate matter in a non-urban elementary schoolPLOS ONE

Dear Dr. Schiff,

Thank you for submitting your manuscript to PLOS ONE. After careful consideration, we feel that it has merit but does not fully meet PLOS ONE’s publication criteria as it currently stands. Therefore, we invite you to submit a revised version of the manuscript that addresses the points raised during the review process.

ACADEMIC EDITOR: The authors improved the manuscript by addressing most of the reviewers' comments. However, some further corrections are required prior acceptance. I ask the authors to respond in detail to Reviewer 3's comments. 

We look forward to receiving your revised manuscript.

Kind regards,

MARIA LUISA ASTOLFI, Ph.D.

Academic Editor

PLOS ONE

Journal Requirements:

Reviewers' comments:

Reviewer's Responses to Questions

**Comments to the Author**

1. If the authors have adequately addressed your comments raised in a previous round of review and you feel that this manuscript is now acceptable for publication, you may indicate that here to bypass the “Comments to the Author” section, enter your conflict of interest statement in the “Confidential to Editor” section, and submit your "Accept" recommendation.

Reviewer #1: All comments have been addressed

Reviewer #2: All comments have been addressed

Reviewer #3: (No Response)

2. Is the manuscript technically sound, and do the data support the conclusions?

Reviewer #1: Yes

Reviewer #2: Yes

Reviewer #3: Partly

3. Has the statistical analysis been performed appropriately and rigorously? 

Reviewer #1: Yes

Reviewer #2: Yes

Reviewer #3: Yes

4. Have the authors made all data underlying the findings in their manuscript fully available?

Reviewer #1: Yes

Reviewer #2: Yes

Reviewer #3: Yes

5. Is the manuscript presented in an intelligible fashion and written in standard English?

Reviewer #1: Yes

Reviewer #2: Yes

Reviewer #3: Yes

6. Review Comments to the Author

Reviewer #1: The authors have adequately addressed the previous comments. Relevant results are stated in the discussion section, not in the conclusion section.

Reviewer #2: I appreciate authors' efforts to address all the comments that I made. I have no further comment and thus believe that the current manuscript is ready for publication.

Reviewer #3: Dear Authors,

Unfortunately, my points were not fully considered.

Noise level: A sound pressure level of 60 dB(A) in a school is totally unacceptable. The American National Standards Institute recommends a maximum level of 35 dB(A) in classrooms, the same applies to the guidelines in Europe, most of which also consider a maximum value of 35 dB(A) to be acceptable, although maximum values of up to 40 dB(A) can be found in guidelines.

I consider this point critical for classrooms, since the use of such devices requires the acceptance of teachers and students. Especially in classrooms, guidelines on noise levels should be discussed.

The measurement setup is not yet sufficiently described.

Nevertheless, I find the approach of this work good and the investigations carried out interesting. The presentation of the results is also very good. The description of the measurements and their structure, which would be important for a deeper understanding and classification, is still missing.

7. PLOS authors have the option to publish the peer review history of their article (what does this mean?). If published, this will include your full peer review and any attached files.

Reviewer #1: No

Reviewer #2: No

Reviewer #3: No

---

## [Author Response · Author response to Decision Letter 1]

21 Oct 2022

Response to Editors’ Comments (October 2022)

*** Please see page 6 for the previous responses to the reviewers’ comments that were addressed in the first round of revisions and were accepted and see page 2-3 (response highlighted in green) for our response to the reviewers’ comment that needed to be revised after our prior submission 

Comments to the Author

1. If the authors have adequately addressed your comments raised in a previous round of review and you feel that this manuscript is now acceptable for publication, you may indicate that here to bypass the “Comments to the Author” section, enter your conflict of interest statement in the “Confidential to Editor” section, and submit your "Accept" recommendation.

Reviewer #1: All comments have been addressed

Reviewer #2: All comments have been addressed

Reviewer #3: (No Response)

2. Is the manuscript technically sound, and do the data support the conclusions?

Reviewer #1: Yes

Reviewer #2: Yes

Reviewer #3: Partly

3. Has the statistical analysis been performed appropriately and rigorously? 

Reviewer #1: Yes

Reviewer #2: Yes

Reviewer #3: Yes

4. Have the authors made all data underlying the findings in their manuscript fully available?

Reviewer #1: Yes

Reviewer #2: Yes

Reviewer #3: Yes

5. Is the manuscript presented in an intelligible fashion and written in standard English?

Reviewer #1: Yes

Reviewer #2: Yes

Reviewer #3: Yes

6. Review Comments to the Author

Reviewer #1: The authors have adequately addressed the previous comments. Relevant results are stated in the discussion section, not in the conclusion section.

Reviewer #2: I appreciate authors' efforts to address all the comments that I made. I have no further comment and thus believe that the current manuscript is ready for publication.

Reviewer #3: Dear Authors,

Unfortunately, my points were not fully considered.

Noise level: A sound pressure level of 60 dB(A) in a school is totally unacceptable. The American National Standards Institute recommends a maximum level of 35 dB(A) in classrooms, the same applies to the guidelines in Europe, most of which also consider a maximum value of 35 dB(A) to be acceptable, although maximum values of up to 40 dB(A) can be found in guidelines.

I consider this point critical for classrooms, since the use of such devices requires the acceptance of teachers and students. Especially in classrooms, guidelines on noise levels should be discussed.

The measurement setup is not yet sufficiently described.

Nevertheless, I find the approach of this work good and the investigations carried out interesting. The presentation of the results is also very good. The description of the measurements and their structure, which would be important for a deeper understanding and classification, is still missing.

Response: We added sound level as a limitation in our discussion. We were concerned about the impact of the sound in the classroom; however, the air purifiers and A/C fans were required to be used in the classrooms for health concerns during the COVID-19 pandemic. Teachers were required to have the air purifiers on but teachers had the ability to choose which settings they wanted to use in their classrooms; some used higher settings than others. During our study we varied the treatment settings but these did not include the air purifier setting ‘off” nor did they include the highest air purifier setting. Furthermore, we asked teachers to report on the sound level throughout the study period and to report if the sound level interfered with their ability to teach in the classrooms. We did not receive any reports of noise interfering with the ability to teach in the classroom. Overall, the purpose of the study was to evaluate the impact of various air purifier and fan settings in a classroom setting on particulate matter concentrations and we did not focus on other environmental exposures, although sound is a consideration especially in schools. The use of air purifiers in the classroom has health benefits even though noise can be a concern, however, in our study noise complaints were not received. 

Furthermore, research indicates that 60 dB is conversation volume. According to our research, the CDC notes that 60 dBA is a normal conversation/AC level and that sounds at these levels do not typically cause hearing damage. The CDC further states that 30 dBA is equivalent to a whisper. We also investigated the air purifier used in the classroom and at its maximum level (there are four levels, 1-4) the sound would be 70 dBA, however, in our study the maximum level used was level 3, so less than 70dBA. 70 dBA is the level the CDC notes as causing annoyance by noise, but not hearing damage. We acknowledge that 35 dBA is a recommendation for classrooms, however, it is not a requirement for classrooms and was not realistic in the context of COVID-19 and the requirement for classrooms to use air purifiers. 

We also provided a classroom schematic in the supplementary material, depicting the layout and sizes of the classroom. The schematic also includes dimensions of the classrooms and location of the air purifiers in the classroom in the methods section. Dimensions and description of air purifier and air monitor locations in the classrooms can be found in the methods section of the manuscript. 

CDC Reference: 

https://www.cdc.gov/nceh/hearing_loss/what_noises_cause_hearing_loss.html

Air Purifier Model Technical Specifications (see image below):

https://medifyair.com/collections/air-purifiers/products/ma-112#specs

7. PLOS authors have the option to publish the peer review history of their article (what does this mean?). If published, this will include your full peer review and any attached files.

Do you want your identity to be public for this peer review? For information about this choice, including consent withdrawal, please see our Privacy Policy.

Reviewer #1: No

Reviewer #2: No

Reviewer #3: No

Response to Editors’ Comments (August 2022) 

2. You indicated that ethical approval was not necessary for your study. We understand that the framework for ethical oversight requirements for studies of this type may differ depending on the setting and we would appreciate some further clarification regarding your research. Could you please provide further details on why your study is exempt from the need for approval and confirmation from your institutional review board or research ethics committee (e.g., in the form of a letter or email correspondence) that ethics review was not necessary for this study? Please include a copy of the correspondence as an ""Other"" file.

Furthermore, we recommend that the consent statement is revised to indicate that the head teachers at the school provided in parentis loco consent to conduct this study.

Response: Please see the PDF file “IRB Exemption”, categorized as “other”

“Jahred Liddie, a co-author, was supported by a Training Grant in Environmental Epidemiology (T32 E007069) from the National Institute of Environmental Health Sciences. No other specific funding was received for this work by any of the co-authors. Funds from the Department of Environmental Health, Harvard TH Chan School of Public Health and the Harvard Chan NIEHS Center (NIH/NIEHS P30 ES000002) were used to purchase and maintain the equipment used for the study.”

Response: Thank you for this comment. We have added the provided statement to the funding section of the paper and included a Role of Funder statement in the cover letter.

“Jahred Liddie was also supported by a Training Grant in Environmental Epidemiology (T32 E007069) from the National Institute of Environmental Health Sciences. The authors would like to thank Jaime E. Hart and Gary Adamkiewicz for their contributions of expertise and support in this project”

“Jahred Liddie, a co-author, was supported by a Training Grant in Environmental Epidemiology (T32 E007069) from the National Institute of Environmental Health Sciences. No other specific funding was received for this work by any of the co-authors. Funds from the Department of Environmental Health, Harvard TH Chan School of Public Health and the Harvard Chan NIEHS Center (NIH/NIEHS P30 ES000002) were used to purchase and maintain the equipment used for the study.”

Response: These statements were moved to the funding section. 

Response: All the data necessary for the analyses have been uploaded. There are three datasets. 

Comments to the Author

Reviewer #1: Abstract:

The structure of the abstract must be changed. Start with the problem you have found that triggers your research and finalize with the objective, methodology and conclusions.

This is the second sentence of the abstract: “Our objective was to determine if use of air cleaners with HEPA filters and air conditioning (A/C) units were associated with changes in particulate matter (PM) air pollution concentrations in a real-world environment.”

Could you change the order and include the sentence as the main goal of the article? What is the method to achieve that goal?

Response: 

Abstract (Reorganized) 

Given the increased useing of air cleaners in classrooms during the COVID-19 pandemic as a prevention measure, this study aimed to investigate the the effects of portable air cleaners with HEPA filters and window A/C fans on real-time (1 minute) concentrations of PM less than 2.5 microns (PM2.5) or less than 1 microns (PM1.0) in two classrooms in a non-urban elementary school in Rhode Island. For half of each school day, settings were randomized to “high” or “low” for the air cleaner and “on” or “off” for the fan. Descriptive statistics and linear mixed models were used to evaluate the impacts of each set of conditions on PM2.5 and PM1.0 concentrations. The mean half-day concentrations ranged from 3.4 - 4.1 µg/m3 for PM2.5 and 3.4 - 3.9 µg/m3 for PM1.0 On average, use of the fan alone decreased PM2.5 by 0.53 µg/m3 [95% CI: -0.64, -0.42] and use of the filter on high (compared to low) alone decreased PM2.5 by 0.10 µg/m3 [95% CI: -0.20, 0.005]. For PM1.0, use of the fan alone decreased concentrations by 0.18 µg/m3 [95% CI: -0.36, -0.01] and use of the filter on high (compared to low) alone decreased concentrations by 0.38 µg/m3 [95% CI: -0.55, -0.21]. In general, simultaneous use of the fan and filter on high did not result in additional decreases in PM concentrations compared to the simple addition of each appliance’s individual effect estimates. use of either appliance individually. Our study suggests that concurrent or separate use of an A/C fan and air cleaner in non-urban classrooms with low background PM may reduce classroom PM concentrations.

Introduction:

Lines 101-103: “To the best of our knowledge, this is the first study to provide data on the effectiveness of portable air cleaners and fans operating in tandem to reduce PM2.5 and PM1.0 concentrations in occupied classrooms.” This could be the main objective of the article. I suggest not to mention “To the best of our knowledge, this is the first study…” Just state the objective of the article.

Response: The objective of this study is to provide data on the effectiveness of portable air cleaners and fans operating in tandem to reduce PM2.5 and PM1.0 concentrations in occupied classrooms.

Please, unify the format of the tables.

Response: We have adjusted the formatting and file types of the tables to be unified.

Figure 1. Please explain the equation and the parameter “r”.

Response: We have added a caption within the figure describing both these components.

Conclusions:

Use some of the numerical results to reinforce the statements.

Response: Please see the tracked changes document that demonstrate the use of numerical results to reinforce the statements. 

Include some limitations you might have found in the methodology and future improvements.

Response (bold are the changes): This study adds to the limited body of literature surrounding PM2.5 and PM1.0 concentrations in classroom settings in the United States and provides a foundation for future research in the field. Though we did not consider the effects of seasonality given the small sampling period, seasonality should be considered in future studies. Classroom PM concentrations have been shown to vary by season, with higher concentrations of PM occurring in the winter compared to the summer [32]. Occupant behavior (e.g., window opening) would also vary significantly by season. It would also be worthy to more closely evaluate the impacts of occupancy—the number of teachers and students per classroom—on PM concentrations. Multiple studies have shown that mechanical forces are a primary driver of classroom air quality, therefore it is important to look at how the number of students per classroom influences PM concentrations and the effectiveness of different air quality interventions [32]. This study only involved two classrooms as a result of availability of equipment and could have been strengthened if more classrooms were evaluated. As a result, the findings from two classrooms in one school limit the generalizability of the study findings even though the study has multiple day measurements.. Furthermore, to better evaluate the effectiveness of air quality intervention, the air quality device should be placed in the same location and at the height in each classroom to ensure consistency in sampling method. Monitoring the CADR levels at different air quality device operating levels would help contextualize the impact of using the air quality devices at different settings. It is also important to note that this study was conducted in a non-urban area with low ambient PM concentrations and is not reflective of how these air quality interventions may work in urban settings or areas with higher ambient PM concentrations. 

All the data stated in the abstract should have comments in the conclusion section:

“The mean half-day concentrations ranged from 3.4 - 4.1 µg/m 2 for PM 2.5 and 3.4 - 3.9 µg/m 2 for PM 1.0 On average, use of the fan decreased PM 2.5 by 0.53 µg/m 3 [95% CI: -0.64, -0.42] and use of the filter on high (compared to low) decreased PM 2.5 by 0.10 µg/m 3 [95% CI: -0.20, 0.005]. For PM 1.0 , use of the fan decreased concentrations by 0.18 µg/m 3 [95% CI: -0.36, -0.01] and use of the filter on high (compared to low) decreased concentrations by 0.38 µg/m 3 [95% CI: -0.55, -0.21].”

Response: Thank you - we have added this to the conclusions section.

Reviewer #2: 

The manuscript titled ‘Effects of portable air cleaners and A/C unit fans on classroom concentrations of particulate matter in a non-urban elementary school’ documents the effects of air cleaner in the low/high mode and A/C fan in the on/off mode using linear mixed regression models. Interestingly, the manuscript does not discuss potential impacts of students’ activity on classroom PM and on the effects of air cleaner. Because student activity is an important factor affecting classroom PM concentrations and effect of air cleaners on classroom PM, this needs to be discussed in Discussion. The manuscript may also consider the following comments to improve clarity and flow of the manuscript.

Response: Thank you – we have added not being able to record student activity (due to COVID-19 limitations at the time) as a limitation to the study, which can be found in the Discussion. It should be noted that we asked the teachers to track times of events that may impact PM concentrations (e.g. bus arrivals, times during which all students typically leave the classroom), but the data was not sufficient to make any conclusions.

Major comments:

1. Lines 44-47: It should be mentioned that the effect of fan was adjusted for air cleaner; likewise, the effect of air cleaner was adjusted for A/C fan.

Response: Thank you – please see lines 40-46 in the tracked changes document to see the clarification on the adjustments for air clear and the A/C fan. 

2. Lines 48-51: General statement (lines 48-49) of the study findings and the conclusion (lines 50-51) don’t seem to agree in the current writing. It looks like that the conclusion should be modified to more specific one because the effects of concurrent use of an A/C fan and air cleaner on reducing the PM were dependent of the mode of the air cleaner and PM size. The proper conclusion would have implication in saving energy.

Response: Simultaneous use of the fan and air cleaner did not result in significant, additional decreases in PM2.5 - as stated in the results section, the interaction term was actually significantly positive. Regarding PM1.0, no significant differences associated with simultaneous use of the air cleaner and fan were found. We have updated the phrasing of the results in the Abstract as well as in the Results section to better reflect this finding.

3. Lines 48-49: Does the ‘additional decreases in PM concentrations’ mean no interaction effect between fan on and air cleaner on high? But the interaction was only significant for PM2.5 but not for PM1.0. If this statement was from the interaction model outputs, it should be also specific to PM size.

Response: Thank you for this comment. As clarification and as described in the Result section, the interaction term was significantly for PM2.5 and null for PM1.0. This indicated that concurrent use of the appliances, in our study setting, did not result in significantly lower PM concentrations for either size fraction below the simple addition of each separate effect estimates. In line with previous comment, we have rephrased the results description in the Abstract and Result sections to reflect this more clearly.

4. Lines 79-84: The sentences are about health effects of PM, which has nothing to do with use of air cleaner or improved ventilation that seemed to be the theme for the previous sentences within the paragraph. Thus, it would flow better if these were moved to the next paragraph (after the last sentence of the next paragraph that is describing PM health effect).

Response: Given the challenges of installing new HVAC systems or filters in school buildings, portable air cleaners and fans have become popular, less expensive alternatives to reduce PM concentrations. Air cleaners with high-efficiency particulate air (HEPA) filters, which are at least 99.97% efficient at capturing particles 0.3 µm and larger in size, offer the possibility of substantially reducing SARS-CoV-2 viral particles as well as improving overall IAQ [7,5]. Beyond the current context of the COVID-19 pandemic, improved air ventilation is associated with lower school absenteeism, better performance on cognitive function tests, and fewer respiratory symptoms, such as those related to asthma, lung inflammation, and allergies [9, 10, 11]. PM2.5 exposure in children has been associated with asthma incidence (OR=1.10, 95% CI:1.01, 1.20), prevalence of asthma symptoms (OR=1.08, 95% CI:1.02, 1.16), and rhinitis (OR=1.15, 95% CI: 1.05,1.26) [12]. A review study of 33 articles further provided evidence that exposure to particulate air pollution has adverse impacts on children’s respiratory health, with stronger negative effects seen among children with asthma [13]. 

PM is one of the most common pollutants that could potentially degrade air quality in classrooms [14]. Indoor PM levels are influenced by several factors, such as ambient air pollution levels, air exchange rates, occupancy, type and intensity of indoor activities, and particle sizes [14, 15, 16]. One study in Munich, Germany found that PM concentrations in classrooms are about six times higher than outdoor concentrations [17]. Children are particularly vulnerable to potential health consequences related to PM exposure due to their immature respiratory and immune systems and greater breathing rates per body weight [11]. Beyond the current context of the COVID-19 pandemic, improved air ventilation is associated with lower school absenteeism, better performance on cognitive function tests, and fewer respiratory symptoms, such as those related to asthma, lung inflammation, and allergies [9, 10, 11]. PM2.5 exposure in children has been associated with asthma incidence (OR=1.10, 95% CI:1.01, 1.20), prevalence of asthma symptoms (OR=1.08, 95% CI:1.02, 1.16), and rhinitis (OR=1.15, 95% CI: 1.05,1.26) [12]. A review study of 33 articles further provided evidence that exposure to particulate air pollution has adverse impacts on children’s respiratory health, with stronger negative effects seen among children with asthma [13]. 

5. Wondering if the authors examined interaction between classroom humidity and fan or between humidity and air cleaner in the regression models?

Response: We tried this and a model with a three-way interaction - generally, the main effects are almost identical, although there are some significant but small interactions with humidity that vary for the size fractions. However, we decided not to include the result due the difficulty to interpret the impact and the concern that if may distract from results. 

6. Lines 198-205: It seemed that fan effect was bigger for PM2.5 then PM1.0, but air cleaner effect was bigger for PM1.0 than PM2.5. Isn’t this worth to discuss further in Discussion?

Response: Thank you - this point has been addressed in the Discussion.

7. The title of Table 2 should state ‘Unadjusted ANOVA’ if the ANOVA models were not adjusted for anything.

Response: Thank you - we have added this distinction.

8. Lines 235-243: The studies discussed are all home studies. Are there any studies of evaluating the effect of air cleaners in classrooms? Unless discussion is strictly limited to HEPA air cleaners, there are some studies examining effects of air cleaners in classrooms (e.g., Wargocki et al., HVAC R Res. 14 (2008): 327-344; Mattsson and Hygge, 2005 Indoor Air Conference Proceedings, pp 1111-1115; Park et al., Building and Environment 167, 2020: 106437), which should be discussed.

Response: The studies discussed were limited to HEPA air cleaners. We discussed one classroom study that also used HEPA air cleaners, but acknowledged a general lack of existing literature. However, we have cited more school studies elsewhere in the Discussion, including a few mentioned by the reviewer.

9. Lines 256-257: The discussion doesn’t agree with the current study finding. It was reported in Results that the air cleaner on high mode reduced PM1.0 concentrations more than PM2.5. Thus, your argument in this sentence about ‘less effective at removing particles for smaller size PM by air cleaner on high mode’ is not supported by your own finding.

Response: Thank you for this comment. We went to this paragraph and revised our language. We had incorrectly discussed particle size instead of particle concentration since our study was in a classroom that had very low PM concentrations. We also added additional context. Please see lines 365- 398 (in track changes document). 

10. Line 268: ‘….some average daily ambient …’. Specify ‘some’ in this sentence because there are only two days when ambient PM2.5 concentrations were lower than classroom PM2.5.

Response: Thank you - we have added this clarification.

11. Lines 261-273: This paragraph was not clear. What is the main discussion point of the paragraph? The paragraph may need to be modified for clarification.

Response: Thank you – The main discussion of the paragraph was to highlight the fact that the study was conducted in a location that has very low ambient PM concentrations and that those low ambient concentrations may have impacted classroom PM concentrations. We edited the paragraph starting at line 404-422 (in the track changes document) to emphasize the low ambient PM concentrations. 

12. In the limitation section, the small number of classrooms in the study also needs to be mentioned as a limitation. Findings from only two classrooms in one school may not be generalizable although the study has multiple day measurements.

Response (bold are the changes): This study adds to the limited body of literature surrounding PM2.5 and PM1.0 concentrations in classroom settings in the United States and provides a foundation for future research in the field. Though we did not consider the effects of seasonality given the small sampling period, seasonality should be considered in future studies. Classroom PM concentrations have been shown to vary by season, with higher concentrations of PM occurring in the winter compared to the summer [32]. Occupant behavior (e.g., window opening) would also vary significantly by season. It would also be worthy to more closely evaluate the impacts of occupancy—the number of teachers and students per classroom—on PM concentrations. Multiple studies have shown that mechanical forces are a primary driver of classroom air quality, therefore it is important to look at how the number of students per classroom influences PM concentrations and the effectiveness of different air quality interventions [32]. This study only involved two classrooms as a result of availability of equipment and could have been strengthened if more classrooms were evaluated. As a result, the findings from two classrooms in one school limit the generalizability of the study findings even though the study has multiple day measurements.. Furthermore, to better evaluate the effectiveness of air quality intervention, the air quality device should be placed in the same location and at the height in each classroom to ensure consistency in sampling method. Monitoring the CADR levels at different air quality device operating levels would help contextualize the impact of using the air quality devices at different settings. It is also important to note that this study was conducted in a non-urban area with low ambient PM concentrations and is not reflective of how these air quality interventions may work in urban settings or areas with higher ambient PM concentrations. 

Reviewer #3: Dear Author,

Your manuscript follows a very interesting approach. However, important information is missing to be able to assess the results and evaluate them for a school. For example, a room sketch is missing, as well as information about the ceiling height, with the coordinates for the air purifier, A/C unit and also the measuring points. Furthermore, you do not address the CADR values that are realized at different levels of the air cleaner. The sound level at the different levels is also not mentioned, although this is a critical factor for use in classrooms. The same applies to the A/C unit. I read online that a sound pressure level of 57 dB(A) is generated at the "Low" level, which is clearly too high and would in turn have an impact on the students' ability to concentrate. In my view, when you address the specific case in schools, you have to take these points into account.

Response: We added sound level as a limitation in our discussion. We were concerned about the impact of the sound in the classroom; however, the air purifiers and A/C fans were not so loud and did not interfere with the ability to teach in the classroom. Furthermore, internet research indicates that 60 dB is conversation volume. We currently do not have a sketch of the classrooms but we included dimensions of the classrooms and location of the air purifiers in the classroom in the methods section.

---

## [Decision Letter · Decision Letter 2]

9 Nov 2022

Effects of portable air cleaners and A/C unit fans on classroom concentrations of particulate matter in a non-urban elementary school

PONE-D-22-05922R2

Dear Dr. Schiff,

We’re pleased to inform you that your manuscript has been judged scientifically suitable for publication and will be formally accepted for publication once it meets all outstanding technical requirements.

Kind regards,

MARIA LUISA ASTOLFI, Ph.D.

Academic Editor

PLOS ONE

Additional Editor Comments (optional):

All comments have been addressed.

Reviewers' comments:

Reviewer's Responses to Questions

**Comments to the Author**

1. If the authors have adequately addressed your comments raised in a previous round of review and you feel that this manuscript is now acceptable for publication, you may indicate that here to bypass the “Comments to the Author” section, enter your conflict of interest statement in the “Confidential to Editor” section, and submit your "Accept" recommendation.

Reviewer #3: All comments have been addressed

2. Is the manuscript technically sound, and do the data support the conclusions?

Reviewer #3: Yes

3. Has the statistical analysis been performed appropriately and rigorously? 

Reviewer #3: Yes

4. Have the authors made all data underlying the findings in their manuscript fully available?

Reviewer #3: Yes

5. Is the manuscript presented in an intelligible fashion and written in standard English?

Reviewer #3: Yes

6. Review Comments to the Author

Reviewer #3: Dear authors,

Thank you very much for responding in detail to my comments.

I have no further comments.

7. PLOS authors have the option to publish the peer review history of their article (what does this mean?). If published, this will include your full peer review and any attached files.

Reviewer #3: No

---

## [Editor Report · Acceptance letter]

18 Nov 2022

PONE-D-22-05922R2 

Effects of portable air cleaners and A/C unit fans on classroom concentrations of particulate matter in a non-urban elementary school 

Dear Dr. Schiff:

I'm pleased to inform you that your manuscript has been deemed suitable for publication in PLOS ONE. Congratulations! Your manuscript is now with our production department. 

Kind regards, 

on behalf of

Dr. MARIA LUISA ASTOLFI 

Academic Editor

PLOS ONE